

# On the Quasi-2-Day Planetary Waves in the Middle
# Atmosphere During Different QBO Phases
Liang Tang[1], Sheng-Yang Gu[2*], Shu-Yue Zhao[2], Dong Wang[2]
[1] School of Optoelectronic Engineering, Chongqing University of Posts and
Telecommunications, Chongqing, China.
[2] Electronic Information School, Wuhan University, Wuhan, China.
*Corresponding author: Sheng-Yang Gu, (gushengyang@whu.edu.cn)





**Abstract.** We found that the interannual difference of the W3 and W4
Q2DW is significantly correlated with the Quasi-Biennial Oscillation
westerly (QBOW) and easterly (QBOE) phase, identified from the analysis
of the 2003 to 2020 MERRA-2 and SABER atmospheric data. The
amplitude of the zonal wind in the QBOE phase is approximately ~10 m/s
stronger than that in the QBOW phase. Mean zonal easterly winds are
stronger in the QBOE phase than in the QBOW phase, while westerly
winds are stronger in the QBOW phase. The Q2DW is present in the
summer, and the background wind is easterly in both hemispheres. The
mean temperature amplitudes of W3 and W4 in the QBOW phase are
stronger than those in the QBOE phase, and the difference is ~2 K and ~3
K (in the Southern Hemisphere); ~2 K and ~3 K (in the Northern
Hemisphere), respectively. The mean wave period of W4 in the QBOW
phase in the Northern Hemisphere is shorter than that in the QBOE phase.
The W3 mode is modulated by atmospheric eigenmodes in both
hemispheres and shows slight differences in the QBOW and QBOE phases,
while the W4 mode is more likely to show significant differences in the
different QBO phases. Our diagnostic analysis suggests that the
amplification of the QBOW phases W3 and W4 may be due to stronger
mean-flow instabilities and background winds in the mesosphere. In
addition, planetary waves gain stronger source activity during the QBOW
phase to provide sufficient energy for propagation and amplification.



## 1 Introduction

The Quasi-Biennial Oscillation (QBO) is the most prominent feature of the equatorial stratospheric circulation (Holton and Tan, 1980; Holton and Austin, 1991). Since 1953, weakening easterly and westerly winds have been observed in the lower tropical stratosphere for approximately 28 months (Baldwin et al., 2001; Naujokat, 1986; Veryard and Ebdon, 1961; Reed et al., 1961). The QBO is sustained mainly by large-scale Kelvin waves, Rossby gravity, gravity waves, and momentum deposits from vertical advection. The latitude range of QBO observation is 15˚N-15˚S. The influence of QBO on the tropical stratosphere and the tropical troposphere has been well known, for example, the Hadley circulation (Gray et al., 1992; Zhang et al., 2022; Jiang et al., 2022; Sun et al., 2022; Reid et al., 2022) and the Madden–Julian Oscillation (Yoo and Son, 2016; Son et al., 2017; Marshall et al., 2017; Nishimoto and Yoden, 2017; Hood et al., 2020; Martin et al., 2019). Nao et al. (2010) and Yamashita et al. (2011) indicate the secondary circulation caused by equatorial QBO is important in the middle stratosphere, but not in the lower stratosphere. Peña-Ortiz et al. (2019) studied the effect of QBO on tropical convection and revealed the regulating effect of QBO on tropical convection. They found that tropical convection influences stationary waves and polar vortices in the southern hemisphere during winter. Lu et al. (2014) and Garfinkel et al. (2012) consider the significance of the meridional

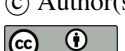



circulation anomaly caused by QBO extending from subtropical to mid-latitude through the change of refractive index and the modulation of Rossby wave propagation.

The QBO impacts the extratropical stratosphere by modulating the strength of the stratospheric polar vortex (Gray et al., 2004; Baldwin et al., 2001; Holton and Austin, 1991; Holton and Tan, 1980). Tian et al. (2019) believe that QBO has a greater influence on the interannual tropical water vapor anomalies than El Niño–Southern Oscillation (ENSO) in the middle and lower stratosphere. For the westerly phase of the QBO (QBOW), the tropical water vapor interannual anomaly is positive near the tropopause and in the lower stratosphere, negative in the middle stratosphere, and positive in the upper stratosphere. Vice versa for the easterly phase of the QBO (QBOE). Ma et al. (2021) found that the East Asian winter monsoon in the early winter months is weaker in the QBOE than in the QBOW during 1958–2019. In addition, they found that the activity of planetary waves also changes in association with the QBO. Their examination of the zonal wavenumbers (WNs) of planetary waves in the sea level pressure (SLP) field shows that WN1 strengthened and WN 2 and WN 3 weakened during QBOE.

Yamazaki et al. (2020) propose that the tropospheric anomaly generates a Rossby wave train that propagates into the mid-latitude troposphere and interferes with stationary waves, especially with



wavenumber 1, resulting in enhanced upward planetary wave propagation
and weakened polar vortex. Based on the modulation of the 11-year solar
cycle, they concluded that the QBO (QBOE-QBOW) in outgoing
longwave radiation (OLR) was strong in the solar min years with
significantly enhanced convection over the western tropical Pacific. Li et
al. (2020) demonstrated for the first time that the synoptic Rossby waves
enhanced at ~40 hPa in the tropics in February 2016 came from both the
extratropical and the local wave generation. They suggest that the forcing
of the unusually long-lasting westerly zonal phase in the middle
stratosphere (~20 hPa), is mainly caused by the enhanced Kelvin wave
activity. Lu et al. (2019) found that the interannual variation of ~2-5 days
eastward propagating planetary waves was positively correlated with
zonal-mean zonal winds averaged over $67.5°±10°S$, but negatively
correlated with the QBO index in the southern winter. In addition, they
believe that the growth rate (stronger wave) of eastward propagating
planetary waves generated by strong polar night jet (PNJ) is greater in
QBOE than in QBOW phase, explaining the QBO-like signal in Antarctic
planetary waves.

The short-period (~2–16 days) planetary Rossby waves propagate

westward relative to the ground, which is primarily caused by the uneven
distribution of sea-land topography and atmospheric temperature.
Planetary waves cause significant perturbations and diurnal variability in





the dynamics, chemistry, and composition of the mid-latitude stratosphere
and mesosphere (Qin et al., 2021b; Qin et al., 2021a; Iimura et al., 2021;
Liu et al., 2020; Liu et al., 2019; Xiong et al., 2018; Pancheva et al., 2018).
Quasi-2-day waves (Q2DWs) play an important role in planetary wave
observation because of their shorter period and stronger amplitude (Iimura
et al., 2021; Gu et al., 2021; Gu et al., 2019; Pancheva et al., 2018; Kumar
et al., 2018; Gu et al., 2018; Ma et al., 2017; Pancheva et al., 2016).
Previous literature studies mainly focused on the observation of Q2DW
with zonal wavenumbers 2 (W2), 3 (W3), and 4 (W4). The amplitude of
Q2DW W3 is the strongest among the three modes in the Southern
Hemisphere (SH), and W4 is relatively strong in the Northern Hemisphere
(NH). The amplitude of W2 is relatively weaker than that of W3 and W4
in both hemispheres, but it can nevertheless be measured by space-based
detectors. The propagation and amplification of Q2DWs are closely related
to a theory of normal modes. (Salby, 1981a; Salby, 1981b) believed that
W3 and W4 were Rossby-gravity wave modes $(3, 0)$ and $(4, 0)$, respectively,
in the real atmosphere, which could modulate and extract energy through
the background mean flow. However, Plumb et al. (1983) suggested that
the amplification and propagation of Q2DWs might be the result of the
barotropic/baroclinic instability in the middle-high latitudes of the summer
hemisphere. Previous studies have extensively discussed the mechanism of
dual propagation and amplification of Q2DWs. Q2DWs are usually





defined as having the characteristics of normal modes while being
amplified due to barotropic/baroclinic instabilities.
The WACCM + data assimilation research method was proposed by
(Gu et al., 2016). They found that the largest Q2DWs amplitudes in the SH
during 2007 occurred in early Jan (W2), late Jan (W3), and late Feb (W4),
respectively, and indicated that background conditions in these three
periods favored the propagation and amplification of these three modes.
Mccormack et al. (2014) used the data assimilation system from the
NOGAPS-ALPHA during 2007-2009 to find the changes between Q2DWs
and migrating diurnal tides. In addition, the short-term variation of
migrating diurnal tides may be caused by the nonlinear interaction between
tidal waves and Q2DWs (Chang et al., 2011). Gu et al. (2021) found that
W3 and W4 occurred more frequently around 48 h, while W2 tended to be
short-period events. In addition, they found that the wave periods of W3
and W4 rise during the late summer in the Northern Hemisphere. Tang et
al. (2021) found the eastward wave in winter periods and westward
background wind in both hemispheres. In addition, they observed that the
mean phase speeds of zonal wavenumbers -1 (E1), -2 (E2), -3 (E3), and -4
(E4) were relatively stable, which are ~53 m/s, ~58 m/s, ~55 m/s, and ~52
m/s, respectively, at 70° latitude. Their diagnostic analysis suggests that
mean-flow instabilities in the upper stratosphere and mesosphere may be
responsible for the amplification of PWs.



Previous works have independently investigated anomalous
phenomena of QBO and perturbations of atmospheric circulation and
monsoons, and studies of planetary waves have focused more on variable
wave properties. Few previous studies have explored the relationship
between QBO and planetary waves in detail. Therefore, the present study
focuses on exploring the QBOE/QBOW contributions to W4 and W3
during the 2003-2020 summer.
In this study, we use the Thermosphere, Ionosphere, Mesosphere,
Energetics, and Dynamics (TIMED) satellite and the second modern
retrospective research and application analysis (MERRA-2) datasets to
investigate the contribution of QBO to W3 and W4 westward propagating
waves in the mesosphere during the 2003-2020 summer period.
Specifically, we investigate the distribution of W3 and W4 in QBOE and
QBOW, as well as the variability of planetary waves in different phases;
Also, the role of instabilities, background wind structure, and critical layers
in propagation and amplification. The remainder of the paper is structured
as follows. In Sec. 2, two kinds of datasets and diagnostic analysis and
planetary wave fitting methods used in our study are introduced: the
SABER datasets used to observe Q2DWs, the datasets required for
diagnostic analysis (MERRA2), and the method of diagnostic analysis. In
Sec. 3, we explore the differences between W4 and W3 events during the
2003 to 2022 QBOW and QBOE phases, including the mean temperature



structure, instabilities, and background winds, and reveal the mechanism
of W3 and W4 propagation and amplification in QBOW and QBOE phases.
Sec. 4 provides a summary and conclusions.
**2 Datasets and Analysis Method**
The long-term Sounding of the Atmosphere using Broadband
Emission Radiometry (SABER) observations located on the TIMED
satellite was launched in July 2001 with the mission of studying the human
influence on the least detected and recognized regions of Earth's
atmosphere: the mesosphere, lower thermosphere, and ionosphere (MLTI).
The SABER data set is optimal for studying the variability of Q2DWs, as
they completely cover the vertical region of the mid-atmosphere where the
waves are strongest. The SABER temperature data focuses on a region of
approximately 20 to 120 km, with vertical and latitude resolutions of 4km
and 4°, respectively. In addition, the combined effects of random and
systematic errors in the lower and middle stratosphere are ~1.4 K and ~1
K, respectively (Remsberg et al., 2008). The SABER sampling has two
positions of ~50°S-80°N and ~80°S-50°N, changing every ~60 days. These
data are widely used to study the mid-atmosphere, such as planetary waves,
tidal waves, and climate variability in the mesosphere and lower
thermosphere (Lu et al., 2019; Liu et al., 2019; Gu et al., 2019; Gu et al.,
2018; Huang et al., 2013). These studies suggest that the TMED/SABER
data are plausible and therefore appropriate for our investigation. The

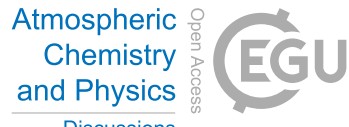

Q2DW events were extracted from the temperature data measured by
SABER from 2002 to 2020, were observed at ~30-40°(S/N) and ~67-73
km for W4, and ~30-40°S and ~79-85 km; ~30-40°N and ~67-73 km for
W3.

The Second Modern Era-Retrospective Analysis for Research and

Applications (MERRA-2) reanalysis data (Gelaro et al., 2017) are used to
provide a background environment from the surface to the top boundary of
~80 km for characterizing the Q2DWs. So far, MERRA-2 produced real
QBO in zonal winds, mean meridional circulation, and ozone (Coy et al.,
2016). The temporal resolution (~1 h), vertical resolution (~2-3 km), and
spatial resolution (~0.5*0.625) of MERRA-2 data were fixed. The
MERRA-2 data have numerous applications in polar atmospheres,
planetary waves, climate change rates, atmospheric circulation, etc. These
studies validate the authenticity of the MERRA-2 data. The long-term
MERRA-2 data from 2003 to 2020 were used to diagnose the contribution
of QBO with different phases to W3 and W4 during the summer and to
analyze background winds, instabilities, source activity, and critical layer
variability.

A least-squares fitting method was used to fit the SABER temperature

observations and extract W3 and W4 for the summer period. In our analysis,
we use a temporal window of ~6 days to analyze the fluctuations of the
period from ~36 to ~60 h in a step of ~2 h. We extract the maximum





amplitude and the corresponding period from the analysis window.
$$y = A\cos[2\pi(\sigma \cdot t + s \cdot \lambda)] + B\sin[2\pi(\sigma \cdot t + s \cdot \lambda)] + C \qquad (1)$$

The amplitude of a planetary wave can be defined as $R = \sqrt{A^2 + B^2}$.
The fitted parameter variables are A, B, and C in Formula (1), respectively.
The planetary wave frequency and zonal wavenumber correspond to $\sigma$
and $s$, respectively. The UT time and longitude of the satellite sample are
$t$ and $\lambda$ respectively. C denotes a zonal mean temperature.
Planetary waves in the lower atmosphere are absorbed or reflected by
the critical layer as they propagate upwards. Planetary waves that gain
sufficient energy in the unstable region are amplified by reflection. In other
words, the critical layer has an important effect on the amplification and
propagation of planetary waves (Liu et al., 2004). In the critical layer, the
phase speed (c) is equal to the background zonal wind ($\bar{u}$).
$$\overline{q_\varphi} = 2\Omega\cos\varphi - \left(\frac{\left(\bar{u}\cos\varphi\right)_\varphi}{a\cos\varphi}\right)_\varphi - \frac{a}{\rho}\left(\frac{f^2}{N^2}\rho\bar{u}_z\right)_z \qquad (2)$$

In the spatial structure of the atmosphere, barotropic/baroclinic
instability may occur in regions with negative latitudinal gradients and
quasigeostrophic potential vorticity ($\overline{q_\varphi}$). Previous studies have shown that
instability contributes to the amplification and propagation of planetary
waves.
In Equation (2), the Coriolis parameter, Earth radius, and background
air density correspond to $f$, $a$ and $\rho$, respectively; $\Omega$ is the angular



speed of the Earth's rotation; $\bar{u}$ is the zonal mean zonal wind; $\phi$ is the
latitude; $N$ is the buoyancy frequency; subscripts $z$ and $\varphi$ are the vertical
and latitudinal gradients.

**3 Results and Discussion**

Figure 1a confirms that the amplitude of mean zonal winds at 40 hPa

above the equator from 2003 to 2020 varies from -20m/s (easterly) to 20
m/s (westerly), obtained from MERRA-2, as shown by periodic changes in
phase with intervals of ~28 months. In Figure 1b, the blue line represents
the QBO index (latitude 0 and 40 hPa), obtained from MERRA-2, the red
shade is the zonal wind -5 m/s to 5 m/s, and $°$ and $*$ represent the W3 and
W4 events. Red for the southern hemisphere and green for the northern.
Annual mean zonal wind variability can be found, with anomalous zonal
wind variability in the QBO in 2010, 2012, 2016, and 2018. The maximum
amplitude of the 2010 zonal wind reaches ~-29 m/s, which is close to the
maximum in the QBOE phase. The amplitude of the zonal wind is
significantly stronger in the QBOE phase than in the QBOW phase,
reaching ~-25 m/s and ~15 m/s, respectively.

Dates of W3 and W4 Q2DW events from 2003 to 2020, obtained with

SABER. The southern hemisphere W3 observations were extracted from
~79-85 km and ~30-40°S; W4 is extracted from ~67-73 km and ~30-40°S.
The northern hemisphere W3 and W4 observations were extracted from
~67-73 km and ~30-40°N. We mainly study the Q2DW difference between





QBOW and QBOE phases, so the following analysis will eliminate the
events with insignificant phase changes (red shading). The W3 and W4
events in the QBOW phase of the Southern Hemisphere summer were
found to be distributed in 2003, 2005, 2007, 2009, 2011, 2014, and 2017.
The W3 and W4 events in the QBOE phase were distributed in 2004, 2006,
2008, 2013, and 2015. Similarly, W3 and W4 events in the QBOW phase
during the Northern Hemisphere summer were distributed in 2004, 2006,
2008, 2009, 2011, 2013, 2017, and 2019. In the QBOE phase, W3 and W4
events are assigned to the years 2003, 2005, 2007, 2010, 2012, 2015, 2016,
2018, and 2020.
Figure 2 shows the seasonal variation of QBOW and QBOE mean
zonal wind at 70°(S/N), obtained from MERRA-2, and the QBOW-QBOE
difference from 2003 to 2020. Figures 2a and 2c show that the background
means zonal winds in the Southern Hemisphere are easterly in summer and
westerly in winter, with the maximum westerly fluctuations being lower
and stronger than the easterly ones. During the QBOW phase, mean zonal
winds in the Southern Hemisphere reach maximum easterly winds of ∼-40
m/s at ~70 km in late December-early January. In August-September,
westerly winds at ~50 km reach ∼75 m/s. Similarly, in the QBOE phase,
the maximum easterly wind at ~70 km reaches ∼-42 m/s from late
December to early January, and the maximum westerly wind at ~50 km
reaches ∼66 m/s from late August to early September. Figure 2e shows the



difference between QBOW and QBOE in the Southern Hemisphere. The
QBOE has stronger easterly winds at 60-80 km than the QBOW in summer,
with a maximum difference of ~3 m/s. Westerly winds are stronger in the
QBOW than in the QBOE at ~40-60 km in winter, with a maximum
difference of ~22 m/s.
Similarly, Figures 2b and 2d show that the background means zonal
winds in the Northern Hemisphere are westerly in winter and easterly in
summer. Mean zonal winds in the QBOW phase reach maximum westerly
winds of ~58 m/s at ~70 km in December and easterly winds of ~-42 m/s
at ~70 km in late June-early July. Similarly, the maximum westerly wind
at ~50 km during the QBOE phase in early January reaches ~46 m/s, and
the maximum easterly wind at ~70 km in July reaches ~-43 m/s. Figure 2f
shows the difference between QBOW and QBOE in the Northern
Hemisphere. QBOW has stronger westerly winds at ~40-80 km in winter
than QBOE, with a maximum difference of ~42 m/s, while QBOE has
stronger westerly winds at ~20-60 km in winter around Jan, with a
difference of ~30 m/s. Easterly winds at ~60-80 km are stronger in the
QBOE than in the QBOW during the summer months, with a maximum
difference of ~2 m/s.
Figure 3 shows the spectral and temperature spatial structure of W3
and W4 in the 2006 and 2007 QBOW and QBOE phases, and the difference
in their spatial structure in the QBOW and QBOE phases. Figure 3A1-3A2



(3B1-3B2) and 3A3-3A4 (3B3-3B4) show the event analysis of the 2006
QBOE and 2007 QBOW phases of W3 (W4) in the Southern Hemisphere.
Figure 3A1 shows the least-squares fitted spectra of SABER temperature
of QBOE phase W3 at ~80 km and ~30-40°S during 13-19 days 2006. The
wavenumber 3 signal with a period of ~43 h is distinctly dominant
throughout the spectrum. Figure 3A2 shows the corresponding temperature
space structure. The temperature spatial structure of W3 reaches a
maximum of ~17 K at ~30-40˚S and ~82 km, and the remaining reaches a
maximum of ~12 K (~68 km). Similarly, Figure. 3A3, the spectrum of W3
in QBOW phase is observed at ~80 km and ~30-40°S on 21-27 days 2007,
when wavenumber 3 becomes the main wave mode and the wave period,
is ~52 h. Figure 3A4 shows that the temperature spatial structure of W3
reaches a maximum of ~14 K at ~30-40˚S and ~82 km. Figure 4A5 shows
that the temperature amplitude of W3 is weaker in the QBOW phase than
in the QBOE phase, with the maximum difference reaching ~7 K. Figure
3B1 shows the observed spectra of W4 in QBOE at ~70 km and ~30-40°S
at 45-51 days 2006, and the wave period of locked wavenumber 4 is ~65
h. Figure 3B2 shows that the temperature spatial structure of W4 is ~2 K
at ~30-40˚S and ~68 km. Similarly, Figure. 3B3, the spectrum of W4 in
QBOW phase is observed at ~70 km and ~30-40°S on 41-47 days 2007,
when wavenumber 3 becomes the main wave mode and wave period is ~49
h. Figure 3B4 shows that the temperature spatial structure of W4 reaches a





maximum of ~4 K at ~30-40˚S and ~68 km. Figure 4B5 shows that the W4
temperature amplitude is stronger in the QBOW phase than in the QBOE
phase, with the maximum difference reaching ~3 K.

Figure 3C1-3C2 (3D1-3D2), and 3C3-3C4 (3D3-3D4), show the 2006

QBOW and 2007 QBOE phase events of W3 (W4) in the Northern
Hemisphere. Figure 3C1 shows the 194-200 days 2006 spectrum of W3 at
~70 km and ~30-40°N in the QBOW phase. The wavenumber 3 signal with
a period of ~47 h is distinctly dominant throughout the spectrum. As
shown in Figure 3C2, the temperature spatial structure of W3 reaches a
maximum value of ~7 K at ~30-40˚N and ~82 km, and another value of ~6
K (~68 km). Similarly, Figure. 3C3, the spectrum of W3 in the QBOE
phase is observed at ~70 km and ~30-40°N on 198-204 days 2007, when
wavenumber 3 becomes the main wave mode and wave period is ~53 h.
Figure 3C4 shows that the temperature spatial structure of W3 reaches a
maximum of ~6 K at ~30-40˚N and ~68 km. Figure 4C5 shows that the W3
temperature amplitude is stronger in the QBOW phase than in the QBOE
phase, with the maximum difference reaching ~2 K. Figure 3D1 shows the
observed spectra of W4 in QBOW at ~70 km and ~30-40°N at 210-216
days 2006, with a wave period of ~47 h for locked wavenumber 4. Figure
3D2 shows that the temperature spatial structure of W4 is ~10 K (~8 K) at
~30-40˚N and ~68 km (~82 km), respectively. Similarly, in Figure 3D3, the
spectrum of W4 in the QBOE phase is observed at ~70 km and ~30-40°N


on 180-186 days 2007, when the wavenumber 4 becomes the main wave
mode and the wave period is ~49 h. Figure 3D4 shows that the temperature
spatial structure of W4 reaches a maximum of ~6 K at ~30-40˚N and ~68
km. Figure 4D5 shows that the W4 temperature amplitude in the QBOW
phase is almost two times stronger than that in the QBOE phase, with the
maximum difference reaching ~7 K.

**3.1 In the Southern Hemisphere**

Spatial structure of mean temperature extracted from events W3 and

W4 in the Southern Hemisphere QBOW and QBOE phases from 2003 to
2020 (see Figure 4). As shown in Figure 4a, the mean temperature spatial
structure of W3 in the QBOW phase has two peak amplitudes at ~68 km
and ~82 km at 30-40˚S, which are ~8 K and ~13 K respectively. The spatial
structure of W3 in the QBOE phase also has a bimodal structure with
maximum fluctuations at ~30-40˚S and ~82 km with an amplitude of ~12
K (Figure 4b). The other peak is at ~68 km, at ~8 K. Figure 4c shows the
difference of W3 in the QBOW and QBOE phases. It is clear that the
temperature amplitude of W3 is stronger in the QBOW phase at ~30-40˚S
and ~70-90 km than in the QBOE phase, and the maximum difference
reaches ~2 K.

Similarly, the mean temperature spatial structure of W4 in the QBOW

phase has two peaks at ~30-40˚S with amplitudes of ~68 km and ~82 km
at ~5 K and ~4 K, respectively (Figure 4d). As shown in Figure 4e, the



fluctuation amplitude of W4 in the QBOE phase is ~2 K at ~30-40˚S and
~70 km (~82 km). Figure 4f shows the difference between W4 in the
QBOW and QBOE phases. It is clear that the temperature amplitude of W4
in the QBOW phase at ~25-50˚S and ~65-90 km is stronger than that in the
QBOE phase, being nearly twice as strong, and the maximum difference
reaches ~3 K.

Figures 5a and 5b show the results of the diagnostic analysis for W3

events in the QBOW and QBOE phases, respectively. W3 in the QBOW is
more favorable for dispersal during the Southern Hemisphere summer,
largely amplified by the mean flow instability between 40-60˚S and ~70-
80 km and the appropriate background winds. In addition, the wave-mean
flow interaction near the critical layer (~48 h) of the green curve is
conducive to the propagation and amplification of W3 (Figure 5a). As
shown in Figure 5b, wave-mean flow interactions of W3 in the QBOE near
the critical layer of the green curve (~47 h) and instability and background
winds in the range of ~40-60˚S and ~70-80 km provide energy for
propagation and amplification. It can be found that the background winds
and instabilities of W3 are stronger in the QBOE phase than in the QBOW
phase. Figures 5c and 5d illustrate the diagnostic analysis of W4 at the
QBOW and QBOE phases, respectively. It is more likely that W4 in
QBOW propagates during the Southern Hemisphere summer, with mean-
flow instabilities and background winds providing sufficient energy to



significantly propagate and amplify W4 at mid-high latitudes and ~70-80
km. In addition, W4 is amplified and propagated by wave-mean flow
interactions near the critical layer (~52 h) of the green curve (Figure 5c).
As shown in Figure 5d, the instability and background wind of W4 in the
QBOE at ~40-60˚S and ~70-80 km, as well as wave-mean flow interactions
near the critical layer of the green curve (~51 h), provide energy for
propagation and amplification. The instability of W3 and the background
wind is weaker in the QBOW phase than in the QBOE phase, which is
inconsistent with the spatial structure result. We suspect that the Southern
Hemisphere W3 mode is mainly affected by the atmospheric eigenmodes
and that the difference between the mean temperature amplitudes in the
QBOW and QBOE phases is small. However, the W4 instabilities and
background winds are stronger in the QBOW phase than in the QBOE
phase, in agreement with the spatial structure results.
**3.2 In the Northern Hemisphere**
Spatial structure of mean temperature extracted from W3 and W4
events during the Northern Hemisphere QBOW and QBOE phases from
2003 to 2020 (see Figure 6). As shown in Figure. 6a, the mean temperature
spatial structure of W3 in the QBOW phase has two peak amplitudes at
~68 km and ~82 km of 30-40˚N, which are ~6 K and ~6 K, respectively.
The spatial structure of W3 in the QBOE phase is unimodal, with the
largest fluctuation at ~30-40˚N and ~68 km, with an amplitude of ~4 K



(Figure 6b). Figure 6c shows the difference between W3 in the QBOW and
QBOE phases. The results show that the temperature amplitude of the
QBOW phase of W3 at ~30-40˚N and ~70-90 km is stronger than that of
the QBOE phase, and the maximum difference reaches ~2 K. As shown in
Figure 6d, the mean temperature spatial structure of W4 in the QBOW
phase has two peaks at ~30-40 ˚N, with amplitudes of ~68 km and ~82 km
at ~7 K and ~4 K, respectively. As shown in Figure 6e, the fluctuation
amplitude of W4 in the QBOE phase is ~5 K (~ 4K) at ~30-40˚N and ~68
km (~82 km). Figure 6f shows the difference between W4 in the QBOW
and QBOE phases. It can be seen that the temperature amplitudes of W4 at
~30-40˚N and ~65-90 km of the QBOW phase are nearly twice stronger as
those of the QBOE phase, and the maximum difference reaches ~3 K.
Figures 7a and 7b show the results of the diagnostic analysis for W3
events in the QBOW and QBOE phases, respectively. W3 in the QBOW is
more conducive to propagation and amplification in the Northern
Hemisphere summer due to mean flow instability between 40 and 60˚N
and 70 to 80 km and appropriate background winds. In addition, the wave-
mean flow interaction near the critical layer of the green curve (~51 h) is
favorable for W3 propagation and amplification (Figure 7a). As shown in
Figure 7b, wave-mean flow interactions of W3 in the QBOE near the
critical layer of the green curve (~51 h) and instability and background
winds in the range of ~40-60˚N and ~70-80 km provide energy for



propagation and amplification. It can be found that the background winds
and instabilities of W3 are stronger in the QBOW phase than in the QBOE
phase. Figures 7c and 7d show the diagnostic analysis of W4 at the QBOW
and QBOE phases, respectively. W4 in QBOW is more likely to propagate
during the Northern Hemisphere summer months, as mean-flow
instabilities and background winds at mid-latitudes and ~70-80 km
provide sufficient energy to significantly enhance W4 propagation and
amplification. In addition, W4 is amplified and propagated by wave-mean
interactions near the critical layer (~43 h) of the green curve (Figure 7c).
As shown in Figure 7d, the instability and background wind of W4 in the
QBOE at ~40-60°N and ~70-80 km, as well as wave-mean flow
interactions near the critical layer of the green curve (~47 h), provide
energy for propagation and amplification. It can be seen that the instability
of W3 and the background wind is stronger in the QBOW phase than in the
QBOE phase, which is consistent with the spatial structure results.
Similarly, the W4 instabilities and background winds are stronger in the
QBOW phase than in the QBOE phase, in agreement with the spatial
structure results. We suspect that the differences between the mean
temperature amplitudes of the W3 and W4 types in the QBOW and QBOE
phases in the Northern Hemisphere are dominated by background
atmospheric instabilities and winds.
**3.3 Comparison between SH and NH**





Figure 8 shows the difference in mean background zonal winds
between W3 and W4 events at the QBOW and QBOE phases in the
Northern and Southern Hemispheres. Figures 8a and 8b show the
difference between W3 and W4 in the Southern Hemisphere. Figure 8a
shows that the mean background zonal wind of W3 at QBOW phase is
weaker than that of QBOE phase on the whole, and the mean background
zonal wind at ~50-70˚S and ~60-80 km (red dashed box) reaches ~8 m/s.
The difference between ~20˚S-20˚N and ~50 km reaches ~35 m/s. The
mean background zonal wind of W4 in the QBOW phase is stronger than
that in the QBOE phase, and the difference between the mean background
zonal wind at ~40-60˚S and ~50-70 km (red dashed box) is ~6 m/s. The
difference between ~20˚S-20˚N and ~50 km reaches ~10 m/s (Figure 8b).
We argue that the zonal winds of W3 in the Southern Hemisphere are
stronger in the QBOE than in the QBOW, but with opposite temperature
amplitudes, because the propagation and amplification of W3 in the
Southern Hemisphere are affected by atmospheric intrinsic models.
However, the zonal winds of W4 in the Southern Hemisphere are stronger
than those of the QBOE in the QBOW, which is consistent with the
structure of the temperature amplitude, suggesting that the propagation and
amplification of W4 in the Southern Hemisphere are susceptible to
atmospheric background variability.
Figures 8c and 8d show the difference between W3 and W4 in the



Northern Hemisphere. Figure 8c shows that the mean background zonal
wind of W3 in the QBOW phase is stronger than that of the QBOE phase
as a whole, and the mean background zonal wind at ~50-70˚N and ~60-80
km (red dashed box) reaches ~6 m/s. The difference between ~20˚S-20˚N
and ~50 km reaches ~11 m/s; The difference between ~5-25˚N and ~60 km
reaches ~11 m/s. The mean background zonal wind of W4 in the QBOW
phase is stronger than that in the QBOE phase, and the difference between
the mean background zonal wind at ~40-60˚N and ~50-70 km (red dashed
box) is ~7 m/s. The difference between ~20˚S-20˚N and ~50 km reaches
~16 m/s (Figure 8d). We believe that the zonal winds of W3 and W4 in the
northern hemisphere are stronger in the QBOW than in the QBOE,
consistent with the temperature amplitude results because the propagation
and amplification of W3 and W4 in the Northern Hemisphere are
susceptible to changes in the atmospheric background.

Figure 9 shows the geopotential height (GPH) amplitudes derived

from the MERRA2 data at 10 hPa for the Southern Hemisphere W3 and
W4 events during the QBOW and QBOE phases, as well as the differences
between the QBOW and QBOE phases. The red line region is the primary
occurrence date of W3 and W4. Figure 9a shows the GPH amplitude of W3
in the QBOW phase, which reaches a maximum amplitude of ~20 m in the
15–30 days source region. The maximum amplitude at the QBOE phase is
~18 m (Figure 9b). As shown in Figure 9c, the difference of W3 in QBOW





and QBOE phases (red dashed box) reaches a maximum of ~11 m. Figure
9d shows the GPH amplitude of W4 in the QBOW phase, which reaches
~9 m in the 15–30 days source region. The maximum amplitude at the
QBOE phase is ~7 m (Figure 9e). As shown in Figure 9f, the difference
between the QBOW and QBOE phases of W4 (red dashed box) reaches a
maximum of ~3 m. We conclude that the source activity in the Southern
Hemisphere of W3 and W4 in the QBOW phase provides stronger energy
than in the QBOE phase to facilitate the propagation and amplification of
W3 and W4 in the Southern Hemisphere.
Figure 10 shows the GPH amplitudes of the W3 and W4 events in the
Northern Hemisphere during the QBOW and QBOE phases, as well as the
difference between the QBOW and QBOE phases. The red line region is
the primary occurrence date of W3 and W4. Figure 10a shows the GPH
amplitude of W3 in the QBOW phase, which reaches ~12 m in the source
region at 195-210 days. The maximum amplitude at the QBOE phase is
~11 m (Figure10b). As shown in Figure 10c, the difference between the
QBOW and QBOE phases of W3 (red dashed box) reaches a maximum of
~3 m. Figure 10d shows the GPH amplitude of W4 in the QBOW phase,
with a maximum amplitude of ~4 m in the source region from 195 to 210
days. The maximum amplitude at the QBOE phase is ~3 m (Figure 10e).
As shown in Figure 10f, the difference between the QBOW and QBOE
phases of W4 (red dashed box) reaches a maximum of ~2 m. We believe





that the easy propagation and amplification of W3 and W4 in the Northern
Hemisphere during the QBOW phase is because the source activity of W3
and W4 is stronger in the QBOW than in the QBOE, providing more energy.
**4 Summary and Conclusions**
We present the first extensive study of the differences between W3
and W4 in the QBOW and QBOE phases, identified from the analysis of
the temperature and wind observations from 2003 to 2020 with SABER
and MERRA-2. We first analyze the differences between the 2006/2007
QBOW and QBOE phases for W3 and W4, since 2006/2007 is
representative of the entire range from 2003 to 2020. W3 and W4 events
from 2003 to 2020 were identified using a two-dimensional least-squares
fit. W4 was observed at ~30-40°(S/N) and ~67-73 km, while W3 was
observed at ~30-40°S and ~79-85 km. W3 was observed at ~30-40°N and
~67-73 km. Our study covers events in both the Northern and Southern
hemispheres and provides a comprehensive diagnostic analysis of their
propagation and amplification in the QBOW and QBOE phases. The main
findings of this study are summarized below:
1. The mean zonal wind amplitude at the equator is stronger in the QBOE
than in the QBOW. Easterly wind amplitudes at 70˚(S/N) and ~70 km (in
early January/middle July) were stronger in QBOE than in the QBOW
phase. Westerly wind amplitudes at 70˚(S/N) and ~70 km (in early
August/late February) were stronger in QBOW than in the QBOE phase.



2. In 2006, the temperature spatial structure of the southern hemisphere W3
in the QBOE phase showed a bimodal structure with amplitudes of ~17 K
and ~12 K at ~68 km and ~82 km, respectively, while the QBOW phase
showed a unimodal structure with a maximum amplitude of ~14 K at ~82
km.
3. The mean temperature amplitudes of W3 and W4 in both hemispheres
are stronger in the QBOW phase than in the QBOE phase. At the same time,
their instabilities and background winds are stronger in the QBOW phase
than in the QBOE phase, except W3 in the Southern Hemisphere.
4. Q2DW is more favorable for propagation in the summer hemisphere of
the QBOW phase, where the mean flow instability and appropriate
background winds in the mid-latitude between 40 km and 80 km
considerably amplify planetary wave propagation. Moreover, the
amplification of planetary waves via wave-means flow interactions can
easily occur near their critical layer.
5. The source activity in the QBOW phase of W3 and W4 in both
hemispheres is more likely to generate sufficient energy to facilitate the
propagation and amplification of W3 and W4 in the QBOW phase than in
the QBOE phase.
Overall, this study reveals a difference between the dynamics of mid-
latitude westward planetary waves in the QBOW and QBOE phases.



*data availability.* MERRA-2 data are available at http://disc.gsfc.nasa.gov.
SABER data were downloaded from http://saber.gats-inc.com/data.php.

Code availability. Code is available at
https://1drv.ms/f/s!AnW2rFlErpPchHIMgX-gOLZGpbXg.

*Author contributions.* LT carried out the data processing and analysis and
wrote the manuscript. SYG, SYZ and DW contributed to reviewing the
article.

*Competing interests.* The authors declare that they have no conflict of
interest.

*Acknowledgements.* This work was performed in the framework of Space
Physics Research (SPR). The authors thank NASA for free online access
to the MERRA-2 temperature reanalysis.

*Financial support.* This research work was supported by the National
Natural Science Foundation of China (41704153, 41874181, and

577 41831071).





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





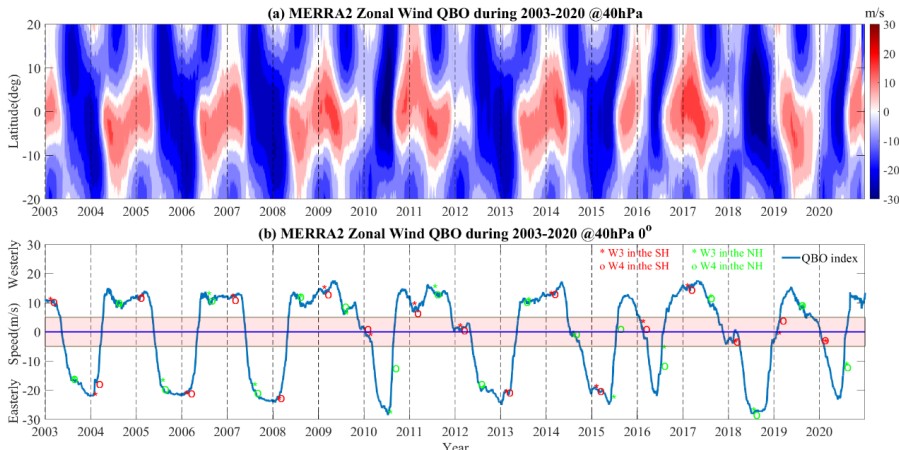

**Figure 1.** MERRA-2 zonal mean zonal wind between 20°S and 20°N at 40 hPa (1a) during 2003-2020. (1b) QBO index (blue line) at 0° and 40 hPa; Dots are the occurrence dates of the maximum amplitudes of W3 (*) and W4 (o). The red/green dots highlight SH/NH. The red-shaded region is the mean zonal wind at -5m/s to 5m/s (1b).

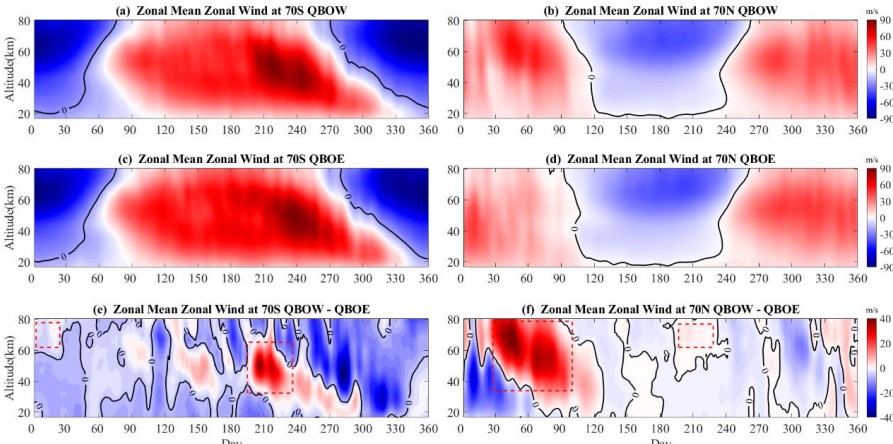

**Figure 2.** Seasonal variations of the zonal mean zonal wind amplitudes in QBOW and QBOE and differences of QBOW-QBOE during 2003-2020 for (a and c) MERRA-2 zonal mean wind in QBOW and QBOE at the 70S,





(b and d) zonal mean wind in QBOW and QBOE at 70N, and (e and f)
differences of zonal mean winds in QBOW to QBOE. The dashed red
boxes in Figures 2e and 2f highlight the regions where the zonal mean wind
enhancement is observed.

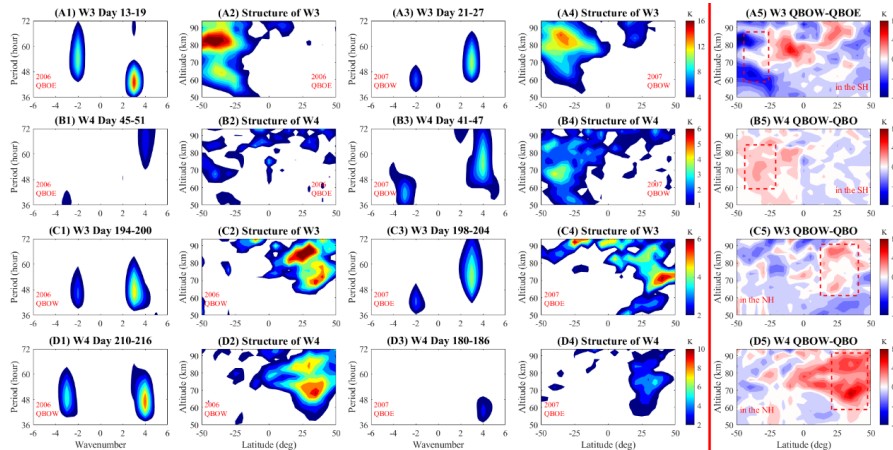


**Figure 3.** The (3A1, 3A3, 3B1, 3B3, 3C1, 3C3, 3D1, and 3D3) spectra and
(3A2, 3A4, 3B2, 3B4, 3C2, 3C4, 3D2, and 3D4) structures of the W3 and
W4 quasi-2-day wave events during 2006/2007 summer period.
Differences of temperature amplitude in QBOW to QBOE (3A-D5). The
2006 SABER temperature observations during days 13–19 (3A1), days 45–
51 (3B1), days 194-200 (3C1), and days 210-216 (3D1) are used; during
2007 days 21-27 (3A3), days 41-47 (3B3), days 198-204 (3C3), and days
180-186 (3D3) are used. Figures 3A1-3A2 and 3B1-3B2 are W3 and W4
of the 2006 summer QBOE phase; Figures 3A3-3A4 and 3B3-3B4 are W3
and W4 of the 2007 summer QBOW phase. Figures 3C1-3C2 and 3D1-
3D2 are W3 and W4 of the 2006 summer QBOW phase; Figures 3C3-3C4
and 3D3-3D4 are W3 and W4 of the 2007 summer QBOE phase. The
dashed red boxes in Figures 3A-D5 highlight the regions where the
temperature amplitude enhancement is observed.

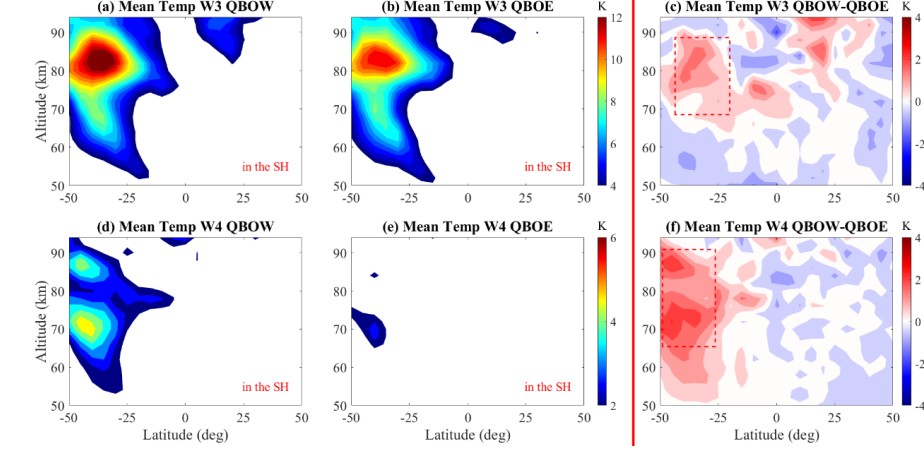


**Figure 4.** The spatial structure of the mean temperatures of the W3 and W4
Q2DWs during the Southern Hemisphere summer is captured in Figures
4a-4b and 4d-4e, respectively. The temperature amplitudes of the W3 and
W4 events in the QBOW and QBOE were extracted from the SABER
temperature data, respectively. Differences of temperature amplitude in
QBOW to QBOE (4c, 4f). The dashed red box highlights the region where
the observed enhancement of the temperature amplitude is observed.

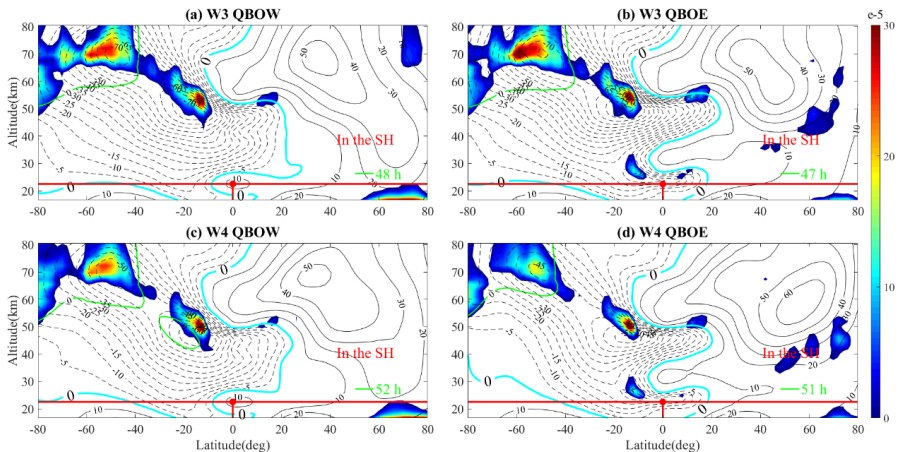

**Figure 5.** Diagnostic analysis results of the QBOW (Figures 5a and 5c) and QBOE (Figures 5b and 5d) quasi-two-day waves for W3 (Figures 5a and 5b) and W4 (Figures 5c and 5d). The blue-shaded region is the instability, the red line is the QBO phase, the cyan line is the null wind, and the green line is the critical layer. The green line shows the critical layer E1 with the mean period.

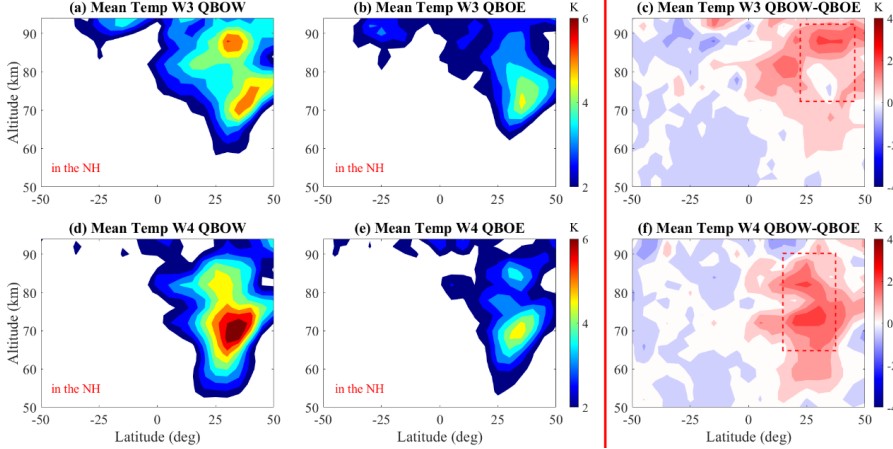

**Figure 6.** Same as Figure 4 but for W3 and W4 during the Northern Hemisphere summer.


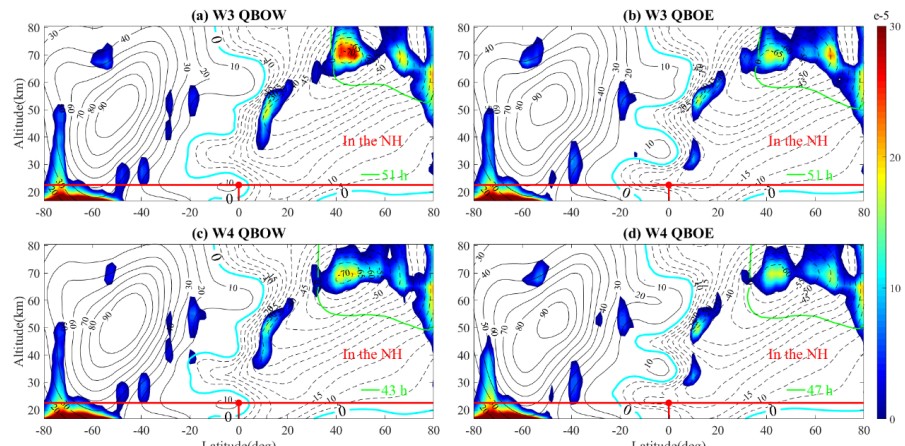

**Figure 7.** Same as Figure 5 but for W3 and W4 during the Northern

Hemisphere summer.

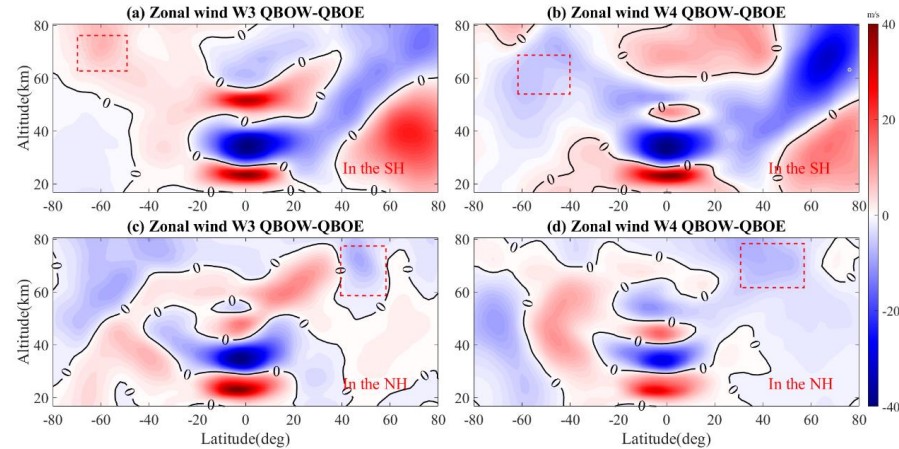

**Figure 8.** Differences in the zonal mean winds from QBOW to QBOE in

SH and NH. (8a, 8c) W3 and (8b, 8d) W4 amplitudes, respectively.

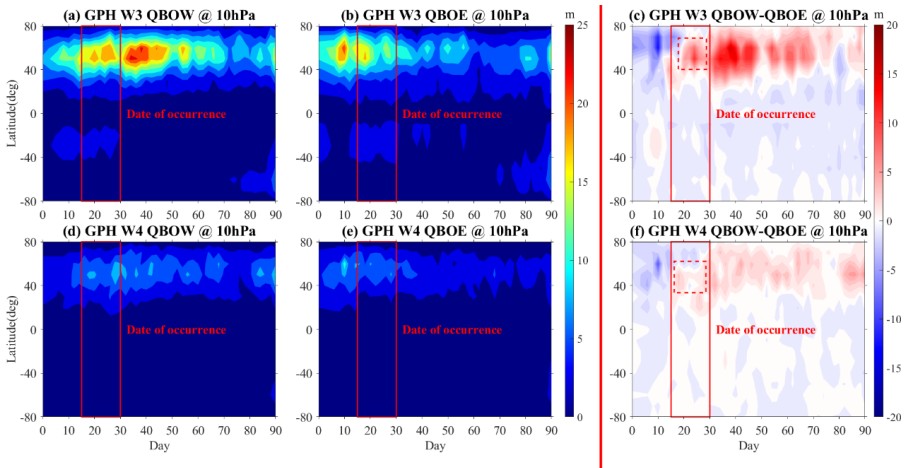

**Figure 9.** Temporal variations of GPH of the QBOW (Figures 9a and 9d) and QBOE (Figures 9b and 9e) for W3 (Figures 9a and 9b) and W4 (Figures 9d and 9e) during Southern Hemisphere summer. Regions enclosed by solid red lines are characterized by the date of major occurrence of W3 and W4. The dashed red box highlights the region where an enhancement of the observed GPH amplitude is observed.

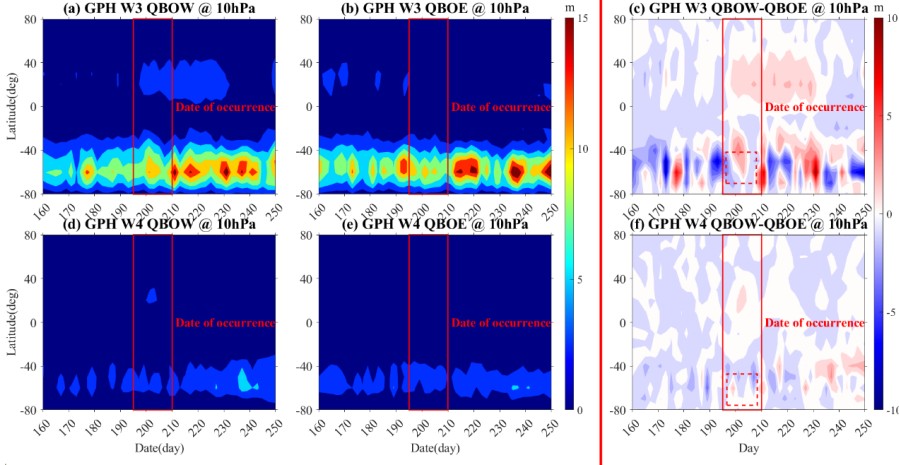

**Figure 10.** Same as Figure 9 but for W3 and W4 during the Northern Hemisphere summer.