# Peer review of "On the Quasi-2-Day Planetary Waves in the Middle"

_Atmospheric Chemistry and Physics, 2022_

## Community Comment (CC3)

The paper suggests non-linear tidal interactions:

*"may be caused by the nonlinear interaction between tidal waves and .."*

I recommend a fundamental shift of how the mechanisms behind the shifts of stratospheric winds is understood. Much of what I will briefly describe is published in **Mathematical Geoenergy** (Wiley/AGU, 2018).

First, we assume that the semi-annual cycle in the upper stratosphere is a result of the nodal cycle about the Earth's ecliptic axis. This results in a semi-annual cycle and not annual cycle due to the symmetry of the hemispheres. In terms of a heuristic mathematical construct, this can be formulated by the multiplication of an annual sinusoidal cycle *convoluted* with a semi-annual delta function impulse train phased according to the (+/-) pairs of solstice or equinox events – a positive (+) for one seasonal event and negative (-) for the complementary event. From this, a rudimentary square wave time-series is generated, with a semi-annual period resulting from the positive excursions pairing to create a positive and similar for the multiplication of the negative excursions. This is adequate to empirically describe the SAO of the upper stratosphere, and identify it with a *forcing* and not a *resonant* condition.

Next, consider that at lower stratospheric altitudes, the QBO cycle of ~28 months takes over. The idea is that a similar nodal construct can be applied but instead of applying only an annual nodal cycling to the convolution, we also add in the nodal lunar tidal cycle. Now we note the important realization that the 0-wavenumber symmetry of the QBO behavior demands that the draconic or nodal lunar cycle of 27.2122 days must be applied to model a *global* effect (not the longitudinally dependent 27.3216 day cycle associated with *regional* tides). This is adequate to empirically describe the QBO of the lower stratosphere, and identify it with a tidal *forcing* where the density is greater and thus more susceptible to gravitational wave energy.

Of course, this hypothesis is completely dependent on the timing of the draconic cycle agreeing with the empirical observation of QBO cycling. The predicted frequency for the multiplication of a draconic cycle *convoluted* with a semi-annual delta function impulse train is calculated as

$$365.25 \text{ modulo } 27.2122 = 0.422 \text{ / year}$$

or 2.368 year period due to physical aliasing of the waveforms (see Mathematical Geoenergy cited above).

This indeed matches well the empirically observed cycling of QBO as shown in the time-series plot below, where all the excursions pair one-to-one with observations, including potentially resolving the issue of the perturbation of 2016.

[Figure]

The question as to why this correlation was missed by atmospheric scientists over the years is difficult to determine. Certainly Richard Lindzen considered the possibility, as the cited quotes below reveal.

*"For oscillations of tidal periods the nature of the forcing is clear"*
Planetary waves on beta-planes, RS Lindzen - Monthly Weather Review, 1967

*"it is unlikely that lunar periods could be produced by anything other than the lunar tidal potential"*
Effects of mean winds and horizontal temperature gradients on solar and lunar semidiurnal tides in the atmosphere
RS Lindzen, S Hong - Journal of the Atmospheric Sciences, 1974

I can only offer again that conventional tidal analysis (for predicting king tides, etc) operates at a local or regional level, where the 27.3216 day lunar synodic (or tropical) cycle is operational. For this particular lunar cycle, modulo arithmetic would generate an aliased ~2.7 year period, which is not close enough to match the long-term QBO periodicity observed. Yet, for conventional tidal analysis, the draconic tidal factor never appears in any analyses, since globally synchronized tides would never be considered, and also importantly, the modulo arithmetic of impulse driven signals at the edge of metastability is also not applied. This means that two critical assumptions – (1) nodal lunar cycling and (2) modulo aliasing – need to be considered, which in retrospect could have easily been overlooked as together they are a necessary condition. A third assumption, that solutions of Laplace's Tidal Equations as applied to the equatorial waveguide can provide the non-linear shaping to allow model fitting to the family of QBO time-series is also described in Mathematical Geoenergy.

The intention of this comment is to provide alternative explanations via geophysics. A tidal approach is much more plausible and parsimonious than attempting to apply variations in solar output via sunspot activity, which the authors of the paper under review consider.  The elegance of transitioning from the semi-annual oscillation of the upper stratosphere to the lunar-modified oscillation of the denser lower density cannot be easily refuted.  If Lindzen had the insight to realize the connection in the late 1960's much more effort could have been applied to model the behavior and apply it to other aspects f climate.

---

## Author Comment (AC1)

We thank the reviewers and editors for their constructive comments on our manuscript. The manuscript is revised thoroughly by considering all the comments. Besides, Figures 1-11 have been updated to make the results clearer. Our responses to every comment are listed below with blue.

**Response to Anonymous Referee 1**

Major comments:

While the title of the manuscript suggests that it is a study of QTDWs, the introduction section may give the impression that the authors are primarily interested in the QBO and its effects on QTDWs. To better reflect the main aim of the study, the authors should revise the introduction section to focus on the investigation of QTDWs and their relationship to different phases of the QBO. In the revised introduction, the authors should clearly state the research question or hypothesis, provide a brief background of the topic, and explain the significance of the study.

The manuscript is revised thoroughly by considering the introduction.

The authors may consider providing a paragraph in the revised introduction briefing on previous works on QTDWs based on ground-based MF and meteor radars, particularly in equatorial regions where the QBO is stronger

More descriptions on the introduction parts are added in the revision. Lilienthal and Jacobi (2015), Gu et al. (2013) and Liu et al. (2019), have been added to the revision.

References

Gu, S.-Y., Li, T., Dou, X., Wang, N.-N., Riggin, D., and Fritts, D.: Long-term observations of the quasi two-day wave by Hawaii MF radar, Journal of Geophysical Research: Space Physics, 118, 7886-7894, https://doi.org/10.1002/2013JA018858, 2013.

Lilienthal, F. and Jacobi, C.: Meteor radar quasi 2-day wave observations over 10 years at Collm (51.3° N, 13.0° E), Atmos. Chem. Phys., 15, 9917-9927, 10.5194/acp-15-9917-2015, 2015.

Liu, G., England, S. L., and Janches, D.: Quasi Two-, Three-, and Six-Day Planetary-Scale Wave Oscillations in the Upper Atmosphere Observed by TIMED/SABER Over ~17 Years During 2002–2018, Journal of Geophysical Research: Space Physics, 124, 9462-9474, https://doi.org/10.1029/2019JA026918, 2019.

The title of the manuscript suggests that QTDWs are observed in the middle atmosphere. However, this term was not explicitly used in the manuscript. Therefore, to avoid confusion, the authors should provide a clear definition of the middle atmosphere in the introduction section. Furthermore, it is important to clarify that the QBO occurs in the stratosphere, while the QTDWs are observed in the mesosphere and lower thermosphere. This clarification will help readers to better understand the vertical location of the phenomena being studied and the relevance of the QBO to the study.

In the methodology section, the authors should explain the process of extracting the wavenumbers and why W3 and W4 are chosen. They should also provide a clear explanation of why W3 and W4 are obtained at different heights.

More descriptions are added in the revision.

The manuscript contains several technical terms, such as W3 and W4, easterly and westerly zonal winds, QBOW and QBOE, NH and SH, and GPH, which may make it challenging for some readers to follow the main focus of the results. To improve the readability and accessibility of the manuscript, the authors should consider defining these terms and providing sufficient context for their use. Additionally, they could consider simplifying the language where possible to help ensure that the main ideas are clearly conveyed.

More concise descriptions on the analysis results are added in the revision.

The authors should highlight the main results of the study, and explain clearly what is new and what is already known. They should also explain the significance and implications of their findings.

The main results of the study are considered, the new results are clearly explained, the significance and implications of the findings are expounded, and the manuscript is thoroughly revised.

The authors should separate the results and discussion sections to make it easier for readers to understand what is already known and what is new. The summary and conclusion section should also provide a clear and concise overview of the main findings of the study and their implications

The results and discussion sections have been described separately, with a clear and concise exposition of the summary and conclusion sections.

By addressing these concerns, the manuscript should become clearer and more accessible to readers.

Minor corrections (not an extensive list, there are many)

Abstract: Please rewrite the abstract, and make sure to mention the most important results.

The abstract has been rewritten in the revison.

Introduction: Please rewrite the introduction, as it currently reads like a small review of QBO. Instead, focus on the QTDW and its variability, and explain why it is important to study it under different phases of QBO. This information is already present in the manuscript, but it needs some re-arrangement.

More descriptions on the introduction parts are added in the revision

Section 2: The authors mention that the SABER data is from 2002 to 2020 (lines:186-187), whereas the MERRA 2 data is from 2003 to 2020. It is recommended that the authors use the same period of data from the two datasets.

Revised in the revision.

Lines 150-151: There is a typo in the expansion of "MERRA 2".

Revised in the revision.

Lines 233-236, Figure 1: The zonal winds vary from negative to positive, not the amplitude. The amplitude cannot be negative.

Revised in the revision.

Lines 246-247: The first sentence of the paragraph is incomplete and needs to be revised

for clarity.

Revised in the revision.

Lines: 247-250: It is important to explain why these specific heights were chosen for W3 and W4, and why different heights were used for each.

Revised in the revision.

Figure 3: The letter 'E' is missing for 'QBOE' in the titles of sub-plots B5, C5, and D5. Additionally, the figure caption should read 'A5-D5,' not '3A-D5' if I understand correctly.

Revised in the revision.

Lines 785-787: Figure 4 caption: The meaning of the sentence "The temperature amplitudes...data, respectively" in the Figure 4 caption is unclear. Please revise the caption to improve clarity. As currently written, it appears that the amplitudes of both W3 and W4 in both QBO phases were extracted from SABER data, but it is unclear if this is correct.

Revised in the revision.

Figure 5: The figure is difficult to understand. It is unclear what the color-scale represents, what its units are, and where the blue shaded region is located. Additionally, it is unclear why one horizontal line is drawn for the QBO phase, and why the green line is mentioned twice, as both the critical layer and critical layer E1 with the mean period are labeled green. The authors should revise the figure and its caption for clarity.

The shaded region represents instability. The green line represents critical layers. More descriptions on the analysis results are added in the revision.

Overall, the manuscript is very confusing and lacks clarity in presenting the important results. The text and figures also contain several typos. Therefore, the manuscript needs to be completely rewritten, with a focus on presenting the results clearly and discussing them in a way that highlights what is new compared to what is already known. Additionally, the introduction should clearly state what the authors want to study, and the manuscript should have a logical flow that leads the reader from the introduction to the conclusions.

We thank the reviewers and editors for their constructive comments on our manuscript. The manuscript is revised thoroughly by considering all the comments.

We thank the reviewers and editors for their constructive comments on our manuscript. The manuscript is revised thoroughly by considering all the comments. Besides, Figures 1-11 have been updated to make the results clearer. Our responses to every comment are listed below with blue.

**Response to Anonymous Referee 2**

This article has some important results, e.g., How the W3 and W4 components of Q2DW in winds and temperature differ during QBOE and QBOW phases. It discussed the role of interaction with mean flow and source variability of planetary waves in modulating the Q2DW variabilities. However, it is not well written and logically organized, which makes it very difficult to interpret and connect one paragraph to the next. It reads like the authors are making sudden jumps from one topic to another without connecting them with previous discussion. There are many flaws on the presentations. Also, many obvious/common knowledge results are presented as if they are new findings.

The manuscript is revised thoroughly by considering all the comments.

Some examples are:

Lines 15-17: "Mean…QBOW phase". This is a characteristic of a QBO and it is very obvious.

Revised in the revision.

Lines 291-292. A sudden and unexpected jump in the description without any motivation or connection to earlier discussion.

We have adjusted the structure of the manuscript to make it easier to understand. Revised in the revision.

No explanation or reasoning is provided on why different days are chosen from different years. For example, line 298, why 13-19 is chosen?

We chose the strongest events of each year, which occurred at different times between 2003 and 2020. Revised in the revision.

Line 367: No reference or description is given to what kind of diagnostic analysis has been performed here. The unit(s) of the diagnostic quantities in Figure 5 are not provided, which makes it difficult to guess it.

The event is analyzed using the method of Equation 2. Revised in the revision.

Lines 485-488: '… GPH W3 amplitude…' What is the meaning of this? It is not clear what this quantity is. Is it some kind of filtered out Q2DW-W3 filtered out from the GPH data?

The fluctuation amplitude of Q2DW in the lower atmosphere was analyzed using GPH data. Revised in the revision.

Line 532: The mean zonal wind amplitude of what? W3 or W4?

Revised in the revision.

A physical explanation is missing: For each of the main finding listed in the summary and conclusion, a valid and proven physical mechanism or explanation should also be

provided.

More concise descriptions on the analysis results are added in the revision.

Minor comments:

Line 10: Define W3 and W4 here. Also, later in the introduction section define what direction they usually propagate.

Revised in the revision.

Line 14: Not clear, what do you mean by amplitude of zonal wind. Is it the zonal-wind due to Q2DW? W3 or W4 or of QBO?

Revised in the revision.

Line 233: same as line 14. 'amplitude of mean zonal wind' – how can wind have an amplitude. Do you mean amplitude of QBO in wind? Make this clear here and in later occurrences.

Revised in the revision.

Line 18: "background wind' define the background wind. Is it ZMZW or wind other than Q2DW-W3 and Q2DW-W4?

Revised in the revision.